# *Lentinus edodes* Polysaccharides Alleviate Acute Lung Injury by Inhibiting Oxidative Stress and Inflammation

**DOI:** 10.3390/molecules27217328

**Published:** 2022-10-28

**Authors:** Yiwen Zhang, Yanfei Cui, Yanbo Feng, Fengping Jiao, Le Jia

**Affiliations:** 1College of Life Science, Shandong Agricultural University, Taian 271018, China; 2School of Public Health, Shandong First Medical University & Shandong Academy of Medical Sciences, Jinan 250117, China

**Keywords:** antioxidant activities, anti-inflammatory, acute lung injury, *Lentinula edodes* polysaccharides, pulmonary protection

## Abstract

Acute lung injury (ALI) is a kind of lung disease with acute dyspnea, pulmonary inflammation, respiratory distress, and non-cardiogenic pulmonary edema, accompanied by the mid- and end-stage characteristics of COVID-19, clinically. It is imperative to find non-toxic natural substances on preventing ALI and its complications. The animal experiments demonstrated that *Lentinus edodes* polysaccharides (PLE) had a potential role in alleviating ALI by inhibiting oxidative stress and inflammation, which was manifested by reducing the levels of serum lung injury indicators (C3, hs-CRP, and GGT), reducing the levels of inflammatory factors (TNF-α, IL-1β, and IL-6), and increasing the activities of antioxidant enzymes (SOD and CAT) in the lung. Furthermore, PLE had the typical characteristics of pyran-type linked by β-type glycosidic linkages. The conclusions indicated that PLE could be used as functional foods and natural drugs in preventing ALI.

## 1. Introduction

As characterized by acute hypoxic respiratory failure, alveolar permeability increase, and severe alveolar edema, acute lung injury (ALI) is complicated lung disease with high mortalities [1]. Since the outbreak of COVID-19 in 2019, there have been more than 500 million clinically confirmed cases globally, including more than six million deaths. The COVID-19 outbreak has resulted in a significant increase in the number of ALI patients [2]. The studies have shown that fever, cough, lung inflammation, and ALI are common symptoms in COVID-19 patients, indicating that ALI can further trigger respiratory failures and acute respiratory distress syndrome, significantly increasing the mortality [3,4]. However, the clinical drugs used to treat ALI, such as dexamethasone and ulinastatin, can cause various adverse effects of coagulation disorder, gastric ulcer, and increased liver burden, greatly limiting the clinical applications in the treatment of COVID-19 patients [5]. Hence, it is urgent to develop novel and non-toxic drugs from natural compounds to treat ALI and its complications.

In recent years, natural compounds extracted from mushrooms have attracted extensive attention because of their good physiological and biological activities, which also laid a foundation for the development of new biopharmaceuticals [6,7,8]. At the same time, *lentinus edodes* as a kind of food, medicine, and functional health products raw material, also attracted people’s attention. *Lentinus edodes*, the first medicinal macrofungus to enter the field of modern biotechnology, is one of the most sought-after edible fungi in East Asia, not only because of its nutritional values, but of its potential for possible therapeutic applications [9]. The report shows that *L. edodes* is rich in nutrients and bioactive substances, such as protein, amino acids, unsaturated fatty acids, vitamins, essential elements such as zinc, iron, calcium, and phosphorus, as well as a large number of metabolites such as polysaccharide and ergosterol, which have high nutritional and health care value [10]. Polysaccharide is the main component and the most important bioactive compound of *L. edodes*. The *L. edodes* polysaccharides (PLE) have been reported to have antioxidant, anticancer, anti-aging, anti-inflammatory, immunomodulatory, antiviral, and other biological activities by interacting with different receptors and triggering downstream signal cascades [11]. Recent studies have shown that the anti-inflammatory and antioxidant mechanism of lentinan is related to NF-κB, PI3K/Akt, MAPK, and JAK/STAT signaling pathways [12]. More interestingly, lentinan has been shown to improve obesity as a dietary supplement. The possible mechanism is that lentinan supplementation upregulates the expression of tight junction protein occludin, reduces the plasma LPS level, and inhibits the accumulation of proinflammatory macrophages in the colon [13]. In addition, lentinan can activate immune cells to fight cancer and tumor through a variety of signaling pathways, such as TLR4/Dectin1-MAPK, Syk-PKC-NFκB and TLR4-PI3K/AKT-NF-κB pathways [14]. Ahn et al. have found that *L. edodes* polysaccharide, as a non-toxic side effect stimulant, can widely regulate the immune system, maturation, proliferation, and immune cell differentiation, obviously affecting in the body’s immune regulation [15]. Therefore, PLE as a biological response regulator has attracted the attention of pharmacologists and clinicians [16,17].

To the best of our knowledge, previous studies mainly focused on the anticancer and immunological effects of PLE. There have been scare reports on the alleviating effects of *lentinus edodes* fruiting body polysaccharides against acute lung injury induced by zymosan. In this work, the PLE was successfully extracted and preliminarily characterized. Eminently, the potential pulmonary protections of PLE were investigated in zymosan-induced ALI mice, aiming to provide valuable guidance for the treatment of ALI and its complications.

## 2. Materials and Methods

### 2.1. Chemicals and Reagents

Fruiting bodies of the artificially cultivated *Lentinus edodes* were collected from the Taian Academy of Agricultural Sciences (Taian, China), identified by Professor Zhang Jianjun of Shandong Agricultural University, and air-dried and ground into a fine powder with a mill before extraction. The diagnostic kits used for investigating the activities of superoxide dismutase (SOD, A001-3-2), catalase (CAT, A007-2-1), and malondialdehyde (MDA, A003-1-2) were purchased from Nanjing Jiancheng Bioengineering Institute (Nanjing, China). The kits for the tumor necrosis factor-α (TNF-α, MB-2868A), inter-leukin-1β (IL-1β, MB-2776A), and interleukin-6 (IL-6, MB-2899A) investigations were supplied by Jiangsu Meibiao Biological Technology Company Limited (Jiangsu, China). Standard monosaccharides including rhamnose (Rha), ribose (Rib), arabinose (Ara), xylose (Xyl), glucose (Glc), fucose (Fuc), mannose (Man), galactose (Gal), glucuronic acid (Glca), and galacturonic acid (Gala) were from Sigma Chemical Company (St. Louis, USA). All the other reagents and chemicals used in the present work were analytical grade provided by local chemical suppliers in China.

### 2.2. Preparation of PLE

The dried *L. edodes* fruiting body residues were crushed into powder using a disintegrator (Shanghai, China). Then the dried powder was extracted by distilled water for 4 h at 90 °C and followed by centrifugation at 3000 rpm for 10 min. The supernatant was concentrated and mixed with 3 times ethanol (95%, *v*/*v*) at 4 °C overnight to obtain precipitation by centrifugation (3000 rpm, 10 min). After deproteinization by the Sevag method and dialysis against water, the PLE was obtained by lyophilization, and used for further analysis [18].

### 2.3. Characterization Analysis

The PLE (1 mg) was mixed with KBr powder (100–200 mg) and then pressed into pellets for infrared spectral analysis measured by a spectrophotometer [19]. The FTIR spectra of the polysaccharide were recorded by a Thermo-Nicolet 6700 FTIR spectrophotometer (Thermo Scientific, Waltham, MA, USA) with a resolution of 4 cm^−1^ and scans of over a range of 4000–500 cm^−1^ at room temperature (23 ± 2 °C). ^1^H and ^13^C NMR measurements were carried out with Bruker AV-300 spectrometer at 300 MHz and 25 °C, with the samples dissolving in deuterated water (D_2_O) [20]. The composition of monosaccharides was analyzed by Ultimate 3000 on the chromatographic column Xtimate C18 (200 mm × 4.6 mm, 5 μm). The monosaccharide composition of PLE was determined by high-performance liquid chromatography (HPLC) reported by Gao et al. with slight modifications [21]. Take each standard sugar and add distilled water to prepare the monosaccharide standard solution with a concentration of 20 μg/mL for use. The polysaccharide was degraded with 2 mol/L trifluoroacetic acid and subjected to pre-column derivatization. The HPLC was applied to analyze derivatives with 0.05 mol/L potassium dihydrogen phosphate solution and acetonitrile solution as mobile phase and detection wavelength at 250 nm. The polysaccharide samples were analyzed by ion chromatography with various mixed and single standards. The monosaccharide species were determined according to the retention time in every single standard. The types and proportions of monosaccharides in polysaccharide samples were determined according to the concentration, retention time, and peak area of mixed standard

### 2.4. Animal Experiments

The Kunming strain mice (male), weighing 20 ± 2 g and purchased from Taian Taibang Biological Products Co., Ltd. (Taian, China), were placed into the animal room under standardized conditions of temperature 22 ± 1 °C, relative humidity 50 ± 10%, and light/dark cycle 12 h with free access to water and standard food [22]. All the experiments were performed in accordance with the Regulations of Experimental Animal Administration issued by the State Committee of Science and Technology of the People’s Republic of China. After adaptation for 7 days, all mice were randomly divided into four groups (10 mice in each group), including normal control (NC) groups, model control (MC) groups, high dose groups (H-PLE), and low dose groups (L-PLE). The mice in H-PLE and L-PLE groups were treated with PLE at doses of 400 or 200 mg/kg/d bw by the gavage procedure, while the mice in NC and MC groups with saline water were used as controls. The whole gavage was continued for 28 successive days. Then, except that in the NC groups with saline water, all mice were intraperitoneally injected with zymosan (0.5 mg/g bw) to successfully induce ALI models as described in the previous report [23]. Another 10 successive interventions were processed after the zymosan injection. The mice were weighed at three time points (initial, intervention, and after modeling). The experiment lasted for 39 consecutive days, and the body weights were investigated. Finally, all mice were sacrificed by exsanguinations under diethyl ether anesthesia after fasting for 12 h.

The characteristic of ALI was investigated by biochemical examination and histological analysis. The serum was separated from the blood samples by centrifugation (14,000 rpm, 10 min), and the gamma-glutamyltransferase (GGT) activities, complement 3 (C3) levels, and hypersensitive C-reactive protein (hs-CRP) levels were analyzed by an automatic biochemical analyzer (ACE, USA). The lung tissue was rapidly excised, homogenized (with PBS (0.2 mol/L, pH 7.4) at 1:9, *w*/*v*), and centrifuged (5000 rpm, 4 °C, 20 min) to gain the supernatants, in order. The pulmonary SOD activities, CAT activities, and MDA contents were analyzed using commercial kits according to the instructions. Additionally, the pulmonary TNF-α, IL-1β, and IL-6 levels were measured using enzyme-linked immunosorbent assay (ELISA) kits. Lung tissue was quickly immersed in a 4% formalin solution for buffer fixation. After rinsing with distilled water, a series of different concentrations of ethanol were used as a dehydrating agent for dehydration treatment. Then xylene was used for transparent treatment and paraffin embedding. Sections of 5 µm were cut using a rotary microtome. Slides were dried in a 55 °C oven for 48 h. The dewaxing and hydration process was completed with xylene and gradient ethanol. The sections were stained with haematoxylin-eosin (H&E) and examined under a light microscope at 200× magnification.

### 2.5. Statistical Analysis

Statistical analysis between groups was analyzed by one-way ANOVA and paired-sample *t*-test (SPSS 16.0 software package, SPSS Inc., Chicago, IL, USA). *p* < 0.05 were considered to be statistically significant.

## 3. Results

### 3.1. Structure Analysis of PLE

As shown in Figure 1, the absorption peak was a typical polysaccharide absorption peak. The strong polysaccharide absorption peak at 3389.68 cm^−1^ was caused by the stretching vibration of -OH, indicating that PLE contained intramolecular hydrogen bonds [24]. The stretching vibration absorption peaks of C-H at 2928.46 cm^−1^ and 1414.99 cm^−1^ were the characteristics of polysaccharides [25]. The absorption peak at 1643.75 cm^−1^ was carbonyl bonds (C=O) and the absorption peaks at 1155.63 cm^−1^, 1080.86 cm^−1,^ and 1023.46 cm^−1^ were the telescopic vibration peak of ether bond (C-O-C), indicating the existence of D-pyran rings [26]. The weak absorption peak at near 900 cm^−1^ demonstrated that the glycosyl residues of PLE were primarily β-type glycosidic linkages [27], showing that the PLE was a typical pyran-type polysaccharide.

As shown in Figure 2A, the ^1^H spectrum contained four anomeric protons at 5.33, 5.12, 4.93, and 4.69 ppm, indicating that the PLE was primarily composed of four types of sugars. The ^1^H NMR spectrum exhibited a set of wide and intense signals (3.0–4.0 ppm) due to the CH_2_-O and CH-O groups. The chemical shifts from 3.2 to 4.1 ppm were assigned to the H-2 to H-6 protons. There was no proton signal at 5.4 ppm, manifesting that PLE was composed of a glucose ring. In ^13^C NMR spectra (Figure 2B), the heterogenous hydrogen was less than δ80 ppm, demonstrating that PLE was pyranose [28]. ^13^C chemical shift focused on 60 to 80 ppm and δ < 5.0 was the β-configuration of pyranose residues in the ^1^H NMR spectrum, demonstrating that the PLE contained a typical β-glucopyranosyl residue, which was in accordance with the analysis of the FT-IR spectrum [29].

As shown in Figure 3, the monosaccharide compositions of PLE were identified by comparing them with the standard sugars. Clearly, the PLE primarily contained four monosaccharides including galacturonic acid (GlcA), mannose (Man), galactose (Gal), and glucose (Glc) in the mass percentages of 2.00%, 1.75%, 2.59% and 93.66% with a molar ratio of 0.35:0.33:0.49:17.63, suggesting that the major monosaccharide in PLE was Glucose.

### 3.2. Effects of PLE on Antioxidant Status

In order to understand the relationship between oxidative stress and zymosan-induced ALI mice, the pulmonary enzyme activities (SOD and CAT) and lipid products (MDA) were determined. As shown in Figure 4, when compared with that in the NC groups, the SOD and CAT activities of mice in MC groups were decreased significantly (*p* < 0.05), indicating that the zymosan administration damaged the antioxidant defense system of lung tissue. However, the SOD and CAT activities of mice in the H-PLE groups were increased by 34.17% and 476.96% (*p* < 0.01) compared with that in the MC groups, respectively. The MDA levels of mice in MC groups (6.35 ± 0.62 μmol/mg prot) were significantly higher than that in NC groups (3.71 ± 0.31 μmol/mg prot, *p* < 0.05), indicating that membrane lipid peroxidation had been triggered. As expected, the MDA levels of mice in the L-PLE and H-PLE groups decreased by 15.59% and 26.46% when compared with that in the MC groups (*p* < 0.05 and *p* < 0.01). These data indicated that PLE expressed dose-dependently superior effects on enhancing antioxidant systems.

### 3.3. Effects of PLE on Inflammation Responses

To uncover the potential anti-inflammatory effects of PLE against ALI-induced inflammations, the pulmonary cytokines levels of TNF-α, IL-1β and IL-6 were investigated. As shown in Figure 5, the TNF-α, IL-1β, and IL-6 levels of mice in MC groups (286.075 ± 3.25 pg/mL, 322.737 ± 2.66 pg/mL, and 29.676 ± 0.76 pg/mL) were increased significantly when compared with that in the NC groups (139.571 ± 1.99 pg/mL, 230.941 ± 2.34 pg/mL and 16.392 ± 0.54 pg/mL), indicating that severe inflammatory reaction had been occurred in lung tissue (*p* < 0.01). Interestingly, the increases could be improved by PLE treatment. The TNF-α, IL-1β and IL-6 levels of mice in H-PLE groups were decreased by 33.78%, 18.27%, and 32.67% when compared with that in the MC groups (*p* < 0.01), suggesting that PLE could reduce inflammatory cytokines and thus inhibit the inflammatory response.

### 3.4. Effects of PLE on Improving Lung Functions

As a classical serum index to evaluate inflammatory lung injury, the GGT activities, C3 levels, and hs-CRP levels were analyzed, and the results were shown in Figure 6. Compared with that in the NC groups, the GGT activity, C3 levels, and hs-CRP levels of that in MC groups were increased significantly (*p* < 0.05), suggesting that the lung function had been destroyed. Interestingly, the GGT activities as well as C3 and hs-CRP levels reached 60.44 ± 1.37 U/L, 142.55 ± 3.26 mg/dL, and 3.39 ± 0.15 ng/L of that in H-PLE groups, with the decreasing rates of 53.74%, 45.93% and 70.95% comparing to that in the MC groups (130.65 ± 2.01 U/L, 263.66 ± 3.64 mg/dL and 11.67 ± 0.27 ng/dL), indicating PLE could significantly recover pulmonic characteristic of ALI.

### 3.5. Histopathological Observations

As shown in Figure 7, after H&E staining, it was found that the lung tissue of mice in NC groups had normal cell morphology, clear cell boundaries, and grid shaped alveoli. However, the lung tissue of mice in MC groups showed pathological properties with incomplete nuclear staining, unclear cell boundaries, infiltration of inflammatory cells and neutrophils, and thickness of alveolar septal edema, indicating the ALI mice suffered serious pulmonary damage. Excitedly, after intervention with PLE, the pulmonary lesions were significantly ameliorated, and the neutrophil accumulation was decreased, suggesting PLE had potential effects in improving lung functions. Besides, PLE showed superior effects at higher dosages.

## 4. Discussions

In recent years, mushroom polysaccharides have been paid more and more academic attention in their applications in treating various diseases focusing on the relationship between bioactivity and structure analysis [30,31]. Modern phytochemical studies show that *L. edodes* is rich in bioactive polysaccharides, especially β-glucan which may play an important role in maintaining its immunobiological activity [32]. Besides, Zhang et al. have reported that *L. edodes* polysaccharides showed higher molecular weights and β-1,6-glucan branch chain, showing significant anti-cancer activity and immunity [33]. In the present work, the characterization results confirmed that PLE was a β-glucan mainly composed of Glc, Gal, Glca, and Man, with the highest content of glucose, indicating that β-glucan may play essential roles in maintaining its pulmonary protective effects.

The present public epidemic of COVID-19 has been attributed to cytokine storms, systemic inflammatory responses, and immune system attacks, causing pneumonia, acute respiratory distress syndrome, and multiple organ failure, and showing the properties of ALI, clinically [34]. Additionally, it has been reported that the SARS-CoV S protein can upregulate IL-6 and TNF-α expressions through the NF-κB signaling pathway, increasing the levels of inflammatory factors in COVID-19 patients, and causing serious lung damage [35,36]. Furthermore, the SARS-CoV-2 virus infection could stimulate the pulmonary release of cytokines and chemokines, which in turn activate immune cells such as macrophages, causing a cytokine storm, excessive oxidative stress response, and a systemic inflammatory cascade, and finally exacerbating the progress of ALI [37,38,39]. Pathogenetically, the TNF-α can accelerate the migration of neutrophils to damaged areas, increasing proteolytic enzyme activities and reactive oxygen species (ROS) levels, and ultimately leading to lung damage [40]. Besides, as a powerful pro-inflammatory cytokine, the IL-1β can activate additional inflammatory cells, accelerating the release of inflammatory mediators, causing a rise of inflammation cascade, amplifying the damage signals, and finally leading to pulmonary edema [41,42]. Furthermore, increased IL-6 may lead to a variety of inflammatory diseases and malignant tumors [43]. Hence, increased levels of TNF-α, IL-1β, and IL-6 were observed by zymosan injection, indicating that a severe inflammatory response had occurred. Altinsoy et al. showed that polysaccharides extracted from wild mushrooms exert anti-inflammatory effects in lipopolysaccharide-stimulated BV-2 cells through the MAPK signaling pathway [44]. Ren et al. have indicated that appropriate lentinan could increase the TNF-α, IL-1β, IL-6, and IL-8 expressions, showing potential anti-inflammatory effects [45]. The results of our current work demonstrate that PLE has a potential therapeutic effect on ALI by inhibiting the inflammatory response.

As the lung appears to be particularly susceptible to oxidative stress and injury, it is critical to accurately evaluate the antioxidant enzymes and lipid peroxidation [46]. Serious oxidative stress, induced by the continuous production of oxygen-free radicals, play a vital role in accelerating the initiation and progress of ALI [47]. The antioxidant enzymes of SOD, CAT, etc., can prevent ROS formation, and convert active oxygen molecules into non-toxic compounds, reducing oxidative stress [48]. The antioxidant mechanism might be that SOD can accelerate the scavenging of superoxide anion radicals to H_2_O_2_, which is harmful to biological structure [49]. Then, the H_2_O_2_ can be decomposed into H_2_O and O_2_ by CAT. Meanwhile, as a reliable indicator of oxidative stress, lipid peroxidation can reflect the extent of tissue damage caused by ROS metabolites [50]. Presently, the zymosan-induced ALI mice showed serious pulmonary oxidative stress, indicating that ALI has suffered serious oxidative stress. As for the antioxidation against ALI compared with other literature, Zi et al. have reported that lentinan exhibits high antioxidant potential, which can reduce the formation of MDA, increase the activity of SOD, and enhance the tolerance of cells to oxidative damage and anti-stress ability [51]. Our results showed that PLE were bioactive antioxidant compounds responsible for alleviating ALI characteristics.

The serum parameters of C3, hs-CRP, and GGT are usually used for the evaluation of lung functions. The alveolar epithelial cells and capillary endothelial cells could be damaged since the AIL occurrence, resulting in the dysfunction of the alveolar air-blood barrier, and then, the GGT could be released from respiratory epithelial cells into the blood, causing a significant serum GGT increase and reflecting the lung injury [52]. Hs-CRP is one of the important sensitive indicators of body inflammation and infection [53]. When the body is invaded by pathogens and stimulated by inflammation, the level of hs-CRP can be increased through the regulation of the TLRs/MyD88 immune pathway [54]. The hs-CRP, as an index reflecting lung inflammation, is an important reference index used in the clinical judgment of infectious diseases [55]. The complement system is an important part of innate immunity, which is mainly involved in the body’s defense against pathogen infection. C3 is the most abundant and important component in the complement system [56]. It is the central link of the two main complement activation pathways and has important biological functions [57]. When the body is under tissue injury and inflammation, the expression of complement C3 increases significantly; conversely, when the tissue injury and inflammation are controlled, C3 can return to normal [58,59]. Our work suggests that PLE inhibits the inflammatory response and protects the lung tissue from injury by decreasing hs-CRP and C3 levels and simultaneously inhibiting GGT activities.

## 5. Conclusions

The present results demonstrated that PLE showed potential effects in relieving ALI and preventing oxidative stress, reflected in the decreasing the serum levels of C3, hs-CRP, and GGT, decreasing the pulmonary levels of TNF-α, IL-1β, and IL-6, increasing the pulmonary activities of SOD and CAT, as well as downregulating the MDA contents, respectively. Furthermore, PLE had the characteristics of a typical pyran-type polysaccharide linked by β-type glycosidic linkages, primarily containing four monosaccharides including galacturonic acid (GlcA), mannose (Man), galactose (Gal) and glucose (Glc). These conclusions indicated that PLE possessed potent antioxidants and anti-inflammation activities and could be used as functional foods and natural drugs in preventing ALI.

## Figures and Tables

**Figure 1 molecules-27-07328-f001:**
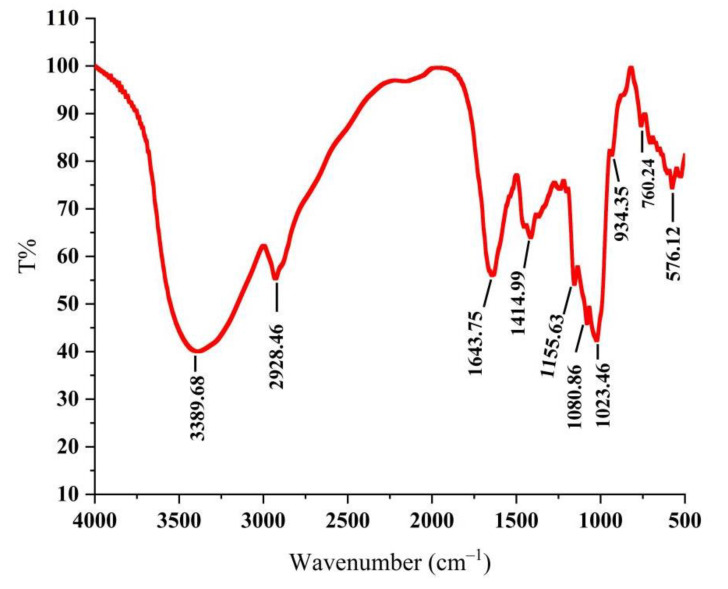
Infrared spectrogram of PLE. PLE: Polysaccharides of *L. edodes*.

**Figure 2 molecules-27-07328-f002:**
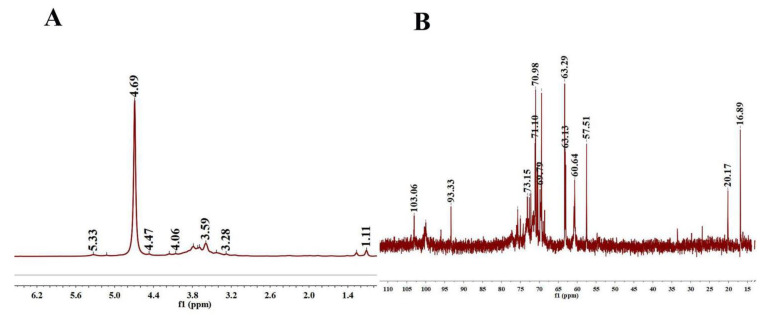
NMR spectra. (**A**) ^1^H-NMR spectra of PLE and (**B**) ^13^C-NMR spectra of PLE. PLE: Polysaccharides of *L. edodes*; NMR: Nuclear magnetic resonance.

**Figure 3 molecules-27-07328-f003:**
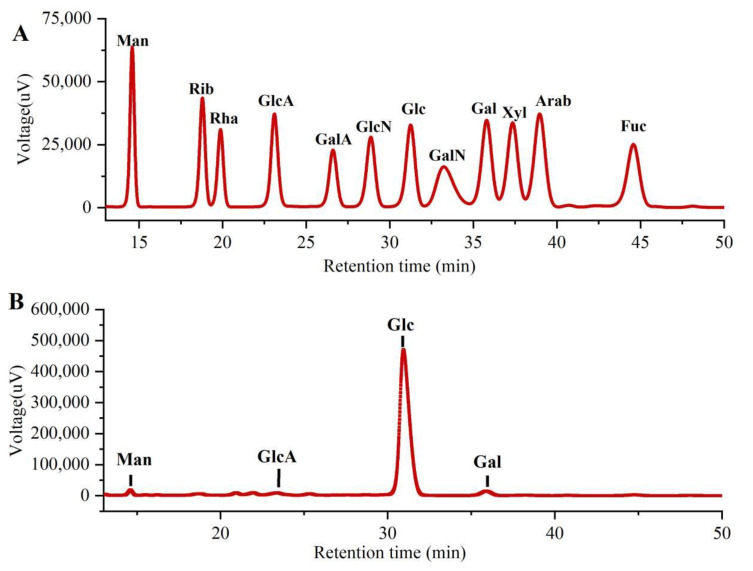
Monosaccharide compositions of PLE by HPLC analysis. (**A**) Standard samples and (**B**) PLE. PLE: Polysaccharides of *L. edodes*. Fuc: fucose; Gal: galactose; GalA: galacturonic acid; GalN: galactosamine; Glc: glucose; GlcA: glucuronic acid; GlcN: glucosamine; Man: mannose; NMR: nuclear magnetic resonance; Rib: ribose; Rha: rhamnose; Xyl: xylose.

**Figure 4 molecules-27-07328-f004:**
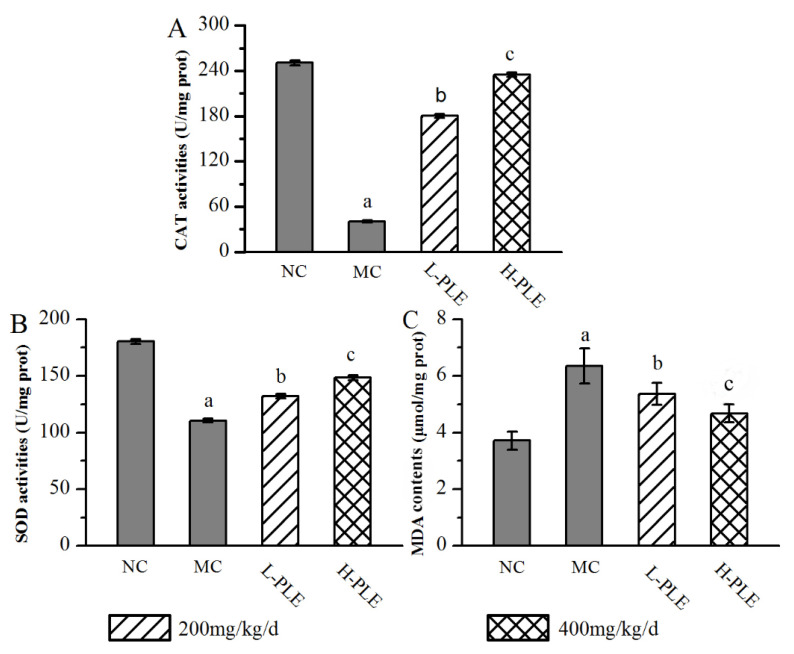
Effects of PLE on antioxidant status in zymosan-induced ALI mice. (**A**) CAT activities, (**B**) SOD activities, and (**C**) MDA contents. The values were reported as the Mean ± SD (n = 10 for each group). (a) *p* < 0.05, compared with NC group; (b) *p* < 0.05, compared with MC group; (c) *p* < 0.01, compared with MC groups. MDA: Malondialdehyde; SOD: Superoxide dismutase; CAT: Catalase; PLE: Polysaccharides of *L. edodes*; NC: Normal control; MC: Model control; L-PLE: low dose group; H-PLE: high dose group.

**Figure 5 molecules-27-07328-f005:**
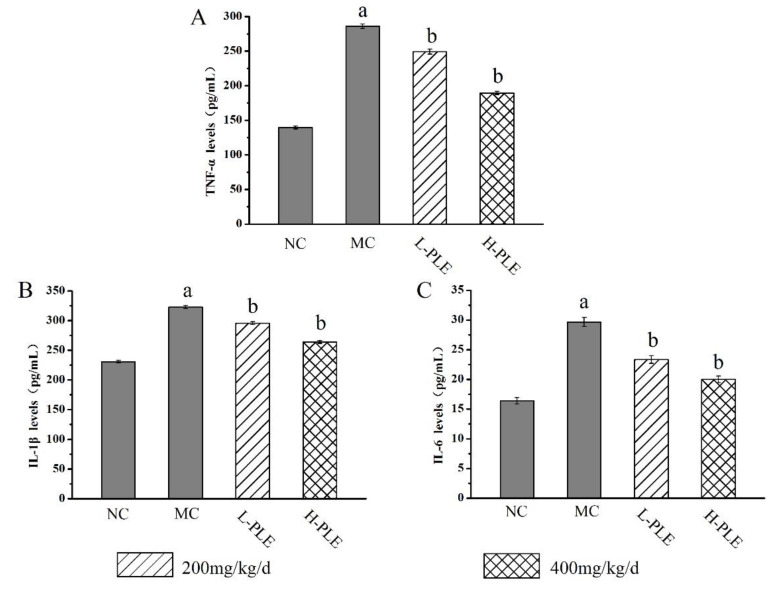
The effects of PLE on inflammation response in zymosan-induced ALI mice. (**A**) TNF-α levels, (**B**) IL-1β levels, and (**C**) IL-6 levels. The values were reported as the Mean ± SD (n = 10 for each group). (a) *p* < 0.01, compared with NC group; (b) *p* < 0.01, compared with MC groups. IL-1β: interleukin-1β; IL-6: interleukin-6; TNF-α: tumor necrosis factor-α. NC: Normal control; MC: Model control; L-PLE: low dose group; H-PLE: high dose group.

**Figure 6 molecules-27-07328-f006:**
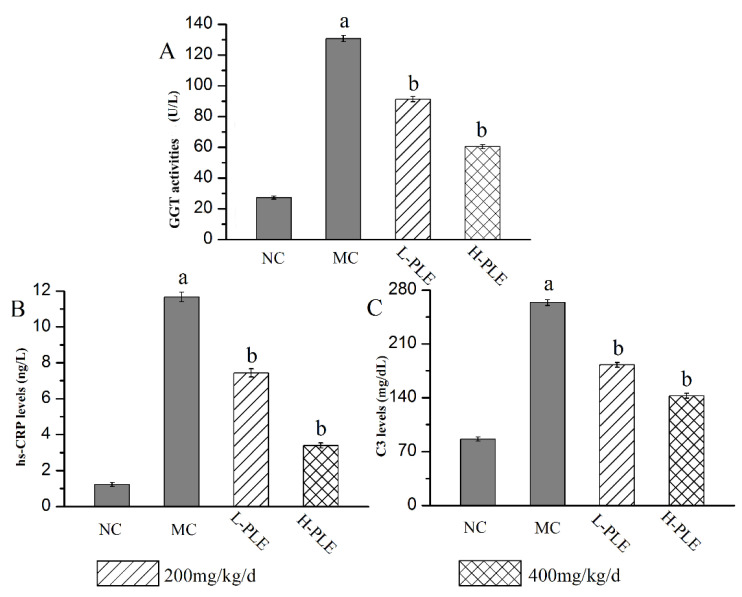
Effects of PLE on improving lung functions. (**A**) GGT activities, (**B**) hs-CRP levels, and (**C**) C3 levels. The values were reported as the Mean ± SD (n = 10 for each group). (a) *p* < 0.05, compared with NC group; (b) *p* < 0.05, compared with MC groups. hs-CRP: hypersensitive C-reactive protein; GGT: Glutamyl transpeptidase. NC: Normal control; MC: Model control; L-PLE: low dose group; H-PLE: high dose group.

**Figure 7 molecules-27-07328-f007:**
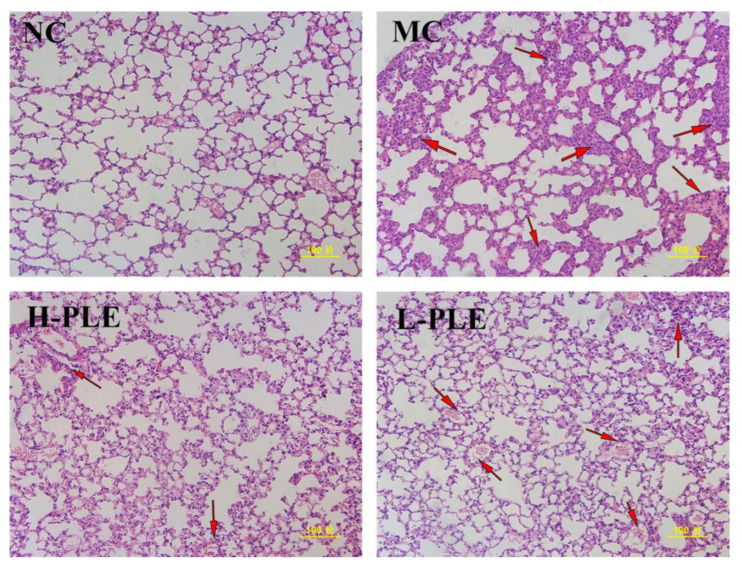
The histopathological observations of the lung by H&E staining (200×). NC: Normal control; MC: Model control; L-PLE: low dose group; H-PLE: high dose group. The red arrows represent neutrophils infiltrating the lung stroma with inflammatory cells and pulmonary telangiectasia with bleeding from cells.

## Data Availability

Data is contained within the article.

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
