# Peer review of "Lentinus edodes Polysaccharides Alleviate Acute Lung Injury by Inhibiting Oxidative Stress and Inflammation"

_molecules, 2022, doi:10.3390/molecules27217328_

Round 1
Reviewer 1 Report
The manuscript entitled “Characterization and pulmonary protective effects of polysaccharides from Lentinula edodes” provides some good results. Therefore, the current manuscript could be accepted for publication, but after going through a minor revision.
1. The title of the manuscript should more attractive.
2. The abstract must be improved by stating the most important outcomes.
3. The authors must explicitly state the novelty of their work at the end of the introduction section.
4. The information of the FTIR instrument must be provided (eg: device name and country of origin).
5. The quality of figure 1 has to be significantly enhanced.
6. How are there two O-H stretching vibrations in the FTIR figure?
7. “The absorption peak at 1643.75 cm-1 was carbonyl bonds (C=O) and the absorption peaks at 1155.63 cm-1, 1080.86 cm-1 and 1023.46 cm-1 were caused by C-O and O-H stretching vibration, indicating the existence of D-pyran rings” Although there are three wavenumbers mentioned in this sentence, there are only two stated vibrations of C-O and O-H stretching!!
8. The quality of figures 2 and 3 has to be significantly improved.
9. Some examples of recently published articles that could be useful for enriching the introduction section and the FTIR discussion are:
https://doi.org/10.1007/s11356-022-21871-x
https://doi.org/10.1016/j.biopha.2020.109946
https://doi.org/10.1016/j.jenvman.2022.115238
https://doi.org/10.1016/j.intimp.2020.106338
10. The conclusion is poorly written, so it has to be improved by stating the most important results.
11. The English language should be improved.
Reviewer 2 Report
point 1. Introduction: the authors mentioned L. edodes polysaccharides protective activities but they do not specify what and how it may act. please specify which kind of receptors or pathways it can modulate
point 2. Materials: please add the catalog number and company of each material used
point 3. Methods:
- the authors should briefly describe how the HPLC method has been performed and the obtained results
- how the authors chose the doses used in the study? please explain the route of administration also
- add the methodology used for histological analysis
- what is the meaning of model control? What is the model used?? The authors mentioned COVID-19 as a positive control, but how did they reproduce the in vivo model? It is very difficult to understand what was the aim of the study
point 4 Results:
- the authors must insert in all Figures a symbol (#, * or other) to clearly demonstrate the significance described in the results paragraph
- figure 7. 200x magnification is really inadequate for affirming the histopathological ameliorations or damage. Consider adding 100x magnification or even less. It is very difficult to identify neutrophils
- to which pathway hs-CRP and C3 belong? It is not clear and defined at all in the results as well as in the discussion sections
